# When Non-Suicidal Self-Injury Predicts Non-Suicidal Self-Injury and Poor Sleep—Results from a Larger Cross-Sectional and Quasi-Longitudinal Study

**DOI:** 10.3390/ijerph182413011

**Published:** 2021-12-09

**Authors:** Habibolah Khazaie, Sepideh Khazaie, Ali Zakiei, Kenneth M. Dürsteler, Annette Beatrix Brühl, Serge Brand, Dena Sadeghi-Bahmani

**Affiliations:** 1Sleep Disorders Research Center, Kermanshah University of Medical Sciences, Kermanshah 6719851451, Iran; hakhazaie@gmail.com (H.K.); zakieiali@gmail.com (A.Z.); bahmanid@stanford.edu (D.S.-B.); 2Psychiatric Clinics, Division of Substance Use Disorders Basel, University of Basel, 4002 Basel, Switzerland; Kenneth.Duersteler@upk.ch; 3Center for Addictive Disorders, Department of Psychiatry, Psychotherapy and Psychosomatics, Psychiatric Hospital, University of Zurich, 8001 Zurich, Switzerland; 4Center of Affective, Stress and Sleep Disorders, Psychiatric Hospital, University of Basel, 4002 Basel, Switzerland; annette.bruehl@upk.ch; 5Substance Abuse Prevention Research Center, Kermanshah University of Medical Sciences, Kermanshah 6719851451, Iran; 6Department of Sport, Exercise and Health, Division of Sport Science and Psychosocial Health, University of Basel, 4052 Basel, Switzerland; 7School of Medicine, Tehran University of Medical Sciences, Tehran 1417466191, Iran; 8Department of Psychology, Stanford University, Stanford, CA 94305, USA

**Keywords:** non-suicidal self-injury, insomnia, sleep quality, nightmares, young adults, prediction

## Abstract

Poor sleep is associated with a higher risk of non-suicidal self-injury (NSSI) as a proxy of unfavorable emotion regulation. In the present study, we tested the hypothesis that past non-suicidal self-injury was associated with current non-suicidal self-injury and with current subjective sleep patterns. To this end, a larger sample of young adults were assessed. A total of 2374 adults (mean age: 27.58 years; 39.6% females) completed a series of self-rating questionnaires covering sociodemographic information, past and current NSSIs, suicide attempts, and current sleep patterns, including experiencing nightmares. Past NSSIs predicted current NSSIs. Current sleep patterns had a modest impact on the association between past and current NSSIs. Compared to male participants, female participants did not report more sleep complaints or more current NSSIs, but more past NSSIs. Past NSSIs predicted the occurrences of nightmares and suicide attempts. The best predictor of current NSSI was the remembered past NSSI, while current poor sleep was only modestly associated with current NSSI. Further indicators of current NSSI and poor sleep were suicide attempts and nightmares within the last six months. Overall, it appears that poor emotion regulation should be considered as underlying factor to trigger and maintain non-suicidal self-injury-related behavior and poor sleep. Further, unlike previous studies, which focused on the possible influence of sleep patterns on NSSIs, the aim of the present study paradigm was to investigate NSSIs on sleep patterns.

## 1. Introduction

Among highly socially discouraged behaviors, non-suicidal self-injury (NSSI) demands special attention due to its high prevalence among adolescents and young adults, and due to its adverse mental long-term consequences for the person with NSSI [1]. More specifically, NSSI is associated with a higher risk of suicide behavior [2]. Non-suicidal self-injury (NSSI) refers to intentional self-inflicted destruction of body tissue without suicidal intention [3,4]. Behavior of NSSI may include cutting, pricking, sandpapering, biting, self-punching, spraying acid, hitting, scratching, and breaking and burning oneself [5], along with the result of the destruction or disfigurement of a part of the body. Further, such behaviors often are repetitive, intentional, and direct, and they are not in line with social expectations and norms [6]. The prevalence rates of NSSI range between 5.5% and 17% in the general population, between 5.5% and 17% among adolescents, 13.4% among young adults, and 5.5% among adults [7]. It follows that the prevalence rate of NSSI is higher during late adolescence and early adulthood, that is to say that NSSIs dramatically increase during the psychosexually, psychosocially, and economically demanding developmental stage of early adulthood. As such, NSSIs are observed in the most economically active population, and the population with the highest pressure to prevail and assert for successful mating [8,9,10].

To explain the emergence and occurrence of NSSIs, at least six highly intertwined dimensions are identified, which for clarity are reported separately. (1) Unfavorable personality traits refer to traits such as low coping skills [1,11], low emotion regulation [1,12,13], high emotion reactivity [14], unfavorable shame-coping [15], and low self-efficacy [14,16]. (2) Psychiatric issues refer for instance to symptoms of panic disorders [13], post-traumatic stress disorders [13], symptoms of anxiety and depression [17,18,19], alexithymia [20,21], and substance use disorders [19], including risky drinking behavior [12]. (3) Past and current social interactions refer to the lack of reliable and long-term relationships (also as a proxy of poor attachment to a reliable person [22]), and the occurrence of childhood maltreatment [1,19], sexual abuse [22,23], and negative life events [24]. (4) Sociodemographic information: younger age [15,19] and female gender [12,19] were associated with higher risks of NSSIs. (5) Ethnic groups refer to the observation that black females are at higher risk, and Asian males are at lower risk to report NSSIs [11] (6) Poor sleep: a recent review [25] identified 16 publications on the relation between poor sleep and the risk of NSSIs. The overall pattern was as follows: poor sleep was associated with NSSIs both cross-sectionally and longitudinally. Further, three out of the 16 publications reported in the review had a longitudinal design: among 392 adolescents, sleep problems at the age of 12–14 years predicted the occurrence of NSSIs two to three years later [26]. Among 881 adolescents, poor sleep predicted the occurrence of NSSI among female adolescents, but not male adolescents [27]. Among 72 university students, those reporting nightmares at baseline reported the occurrence of NSSIs later; negative effects mediated the relationship between nightmares and self-harmful thoughts and behaviors [28]. 

A very similar pattern was observed among 1973 children crossing into adolescence: Participants’ sleep trajectories were observed when children were approximately seven to nine years old, and approximately six years later; three characteristic patterns were observed: those participants who continued to sleeping > 8/night, those moderately decreasing the sleep duration, and those rapidly decreasing the sleep duration. Compared to those continuing to sleep > 8 h/night, those moderately decreasing the sleep duration had a 2.58-fold risk to report NSSIs, and those reporting a rapidly decreasing sleep duration had a 4.86-fold risk to report NSSIs [29]. Given this background, the first aim of the present study was to investigate the associations between subjective sleep patterns and past and current NSSI behaviors. 

Next, while there is a growing body of research on the association between poor sleep and suicidal thoughts and suicidal behavior [30,31], this is not the case regarding the associations between NSSI and nightmares: Among adolescents, a higher prevalence of nightmares was associated with more NSSI and suicidal ideations, both cross-sectionally [32] and longitudinally [33]. We also note that the direction of influence was from nightmares to NSSI and suicidal ideation. However, also the opposite direction is conceivable: a person with NSSI could be impaired in their cognitive-emotional information processing, such to trigger nightmares as a strategy for coping with their concerns and serious mental health issues. With this assumption in mind, the second aim of the present study was to investigate if remembered NSSI occurred 12 months ago predicted the occurrence of nightmares later on. Similarly, the third aim was to associate the occurrence of nightmares with NSSI.

To summarize, a broad variety of psychological, psychosocial, psychiatric, cognitive-emotional, and sleep-related dimensions may explain the emergence and maintenance of NSSIs. In the present study, we focused on the relation between NSSIs and sleep. While there is already extant literature in this regard, we expanded upon the current research in the following four ways. First, we assessed a larger sample of people during early adulthood. As mentioned, this period of psychosocial development as particularly demanding as this age-range reflects the most economically active population, and the population with the highest pressure to prevail and assert for successful mating [8,9,10]. Second, we assessed a community sample, while previous study prevalently focused on high-school or university students. Third, unlike previous studies, which investigated the predictive values of sleep patterns on NSSI, we asked if and to what extent NSSIs could predict sleep issues. Fourth, we designed a quasi-longitudinal study; to this end, participants reported both current NSSIs and NSSIs 12 months before. 

Based on the current research, the following five hypotheses and five research questions were formulated. First, following others [25], we predicted that poor sleep and NSSIs were associated with one another. Second, based on previous research, we predicted that younger age [15,19] was associated with higher NSSIs. We also predicted that compared to their male counterparts, female participants would report more sleep-related issues [34,35] (third hypothesis) and more NSSRIs [12,19] (fourth hypothesis). Next, we assumed that NSSI and the occurrence of nightmares would be associated [32,33] (fifth hypothesis). The explorative research questions were as follows: first, we explored if NSSIs 12 months ago could predict the occurrence of nightmares. Second, we investigated if past NSSIs predicted current NSSIs (second research question), suicide attempts (third research question), and current poor sleep (fourth research question). Last, we explored which factors could predict current sleep quality (Pittsburgh Sleep Quality Index [PSQI], insomnia: Insomnia Severity Index [ISI]) and current NSSIs. To this end, we calculated a series of equations to test the direct and indirect effects of such factors. 

Overall, we claim that the present results are of clinical and practical importance, since data have the potential to identify psychological risk factors of young adults at risk of reporting both past and current NSSIs.

## 2. Methods

### 2.1. Study Design

Young adults aged 18 to 35 years in the Iranian cities of Kermanshah, Sanandaj, Ilam and Hamedan (Western Iran) were approached to participate in this study. To this end, the study was advertised on webpages of health organizations, public hospitals, universities, and private companies. Further, the study was advertised on Social Network Sites (SNS) such as WhatsApp^®^, Instagram^®^, Facebook^®^, telegram^®^, and LinkedIn^®^. Eligible and interested participants received the link to the online survey software of Kermanshah University of Medical Sciences (http://digit.kums.ac.ir). The first page of the online survey provided information about the aims of the study and the secure and anonymized data handling. Thereafter, participants signed the written informed consent; to do so, they ticked the box at the end of the first page to confirm that they understood the aims of the study, the anonymous data handling, and the voluntary study participation. Ticking this specific box also confirmed that participants could withdraw from the study at any time without any further justification. Once a participant agreed with the study conditions, she/he completed a series of questionnaires covering sociodemographic information, and information of non-suicidal self-injury and sleep (see below). Questionnaires were completed in approximately 20–30 min. The study was performed between January and March 2021.

The Ethics Committee of the National Institute for Medical Research Development (NIMAD; Tehran; Iran; code: IR.NIMAD.REC.1399.058) approved the study, which was performed in accordance with the seventh and current version [36] of the Declaration of Helsinki. 

### 2.2. Participants

Inclusion criteria were: (1) age between 18 and 35 years. (2) five or more years of residency in the same city; (3) Iranian nationality; (4) complying with the study conditions; and (5) signed written informed consent. Exclusion criteria were: (1) self-reporting severe mental health issues such as major depressive disorder, bipolar disorders, substance use disorder, personality disorders, anxiety disorders, or adjustment disorders; and (2) self-reporting severe somatic disorders such as obstructive sleep apnea, restless legs syndrome, diabetes, cardiovascular diseases, and neurological diseases such as multiple sclerosis, neuromyelitis optica spectrum disorder, or similar. 

A total of 2598 participants have read the first page of the survey; 2500 (96.2%) ticked the box to agree with the study conditions; 226 (7%) participants either withdraw from study or were identified as “clickthroughs”; the final sample consisted of 2374 (91.4%) participants.

### 2.3. Measures

#### 2.3.1. Sociodemographic Information

Participants reported on their age (years), gender (at birth; female = 1; male = 2), civil status (single = 1; married = 2; divorced/separated/widow = 3), highest educational level (compulsory school = 1; high school diploma = 2; bachelors =3; master’s degree and PhD = 4), and current job position (unemployed = 1; student = 2; employed = 3).

#### 2.3.2. Non-Suicidal Self-Injury; Current

To assess NSSI, we employed the Farsi version of the slightly modified Inventory of Statements about Self-injury (ISAS) [37]. The ISAS a is a self-report questionnaire for measuring the frequency and type of non-suicidal but intentional self-injury. The target behaviors included rubbing skin against rough surfaces, banging/hitting oneself, cutting, biting, hair pulling, burning, needle-sticking, carving, wound picking, pinching, severe scratching, and swallowing chemicals. The key question was: “Within the last four weeks, did you engage in” “carving your skin?”, “burning your skin and/or parts of your body?”, and similar. Answers are forced choice dichotomous answers: yes (=1), and no (=2). A higher sum score reflects a *lower* tendency to NSSI. 

#### 2.3.3. Non-Suicidal Self-Injury; 12 Months Ago

To assess non-suicidal self-injury 12 months ago, we employed the ISAS as described above, though the wording was modified as follows: “About twelve months ago, did you engage in” “carving your skin?”, “burning your skin and/or parts of your body?”, and similar. To improve memory recall, the questionnaire started with the following statements: “Today is [fill in the date; day, month, year]:”. Next: “Twelve months ago, the date was [fill in the date of ONE YEAR AGO; day, month, year]”: Briefly note six key words to describe, what you were doing and how you were feeling ONE YEAR AGO”.

#### 2.3.4. Suicide Attempts

To assess suicide attempts within the last six months, the question was: “Did you try to or seriously think about committing suicide?”. Answers were yes (=1), or no (=2). 

#### 2.3.5. Sleep-Related Information

Subjective sleep: Pittsburgh Sleep Quality Index (PSQI; [38]).

As described elsewhere [39,40], the PSQI is a self-report scale completed in five minutes. It consists of 19 items, and contains seven subscales (subjective sleep quality, sleep latency, sleep duration, sleep efficiency, sleep disturbance, sleeping medication, and daytime dysfunction), each weighted equally on a scale from 0 to 3, with higher scores indicating poorer sleep quality. The seven components are then summed to obtain an overall PSQI score, ranging from 0 (good sleep quality) to 21 (poor sleep quality). Total scores of ≥5 reflect poor sleep, associated with considerable sleep complaints. The Farsi versions of the PDQI have been validated for adults [41], older adults [42], and adolescents [43] (Cronbach’s alpha = 0.85).

#### 2.3.6. Insomnia

Insomnia was assessed using the Insomnia Severity Index (ISI) [44]. The Farsi version has been validated and showed satisfactory psychometric properties [45,46]. The ISI is a self-report questionnaire designed to assess subjective perceptions of sleep complaints. The questionnaire consists of a total of seven items. The first four items relate to difficulty falling asleep and staying asleep, early awakenings, and sleep quality. The remaining items relate to wakefulness and impairment of daily functioning due to sleep complaints. The various items are assigned to scores on a five-point Likert scale (0 = not at all, 5 = very much). The addition of all items gives the total score, which ranges from 0 to 35. Higher scores indicate a higher risk to suffer from insomnia [44].

#### 2.3.7. Nightmares

To assess nightmares, participants answered to the following question: “During the last six months, did you experience nightmares? Answers were: “never” (=1); “rarely” (=2); “sometimes” (=3); “often” (=4); “very often” (=5); and “almost every night” (=6). 

### 2.4. Statistical Analysis

A series of *t*-tests were performed to compare sociodemographic, NSSI-related, and sleep-related information between female and male participants. 

A series of Pearson’s correlations were performed to calculate the associations between, age, current and past NSSIs, current sleep disturbances, and nightmares. 

An Eta-correlation was performed to calculate the associations between past NSSIs and suicide attempts (dichotomous variable).

A multiple regression analysis was performed to predict current NSSI; predictors were age, past NSSI, current sleep complaints, and nightmares. Similarly, two multiple regression analyses were performed to predict sleep complaints (PSQI; ISI). Predictors were age, past and current NSSIs, and nightmares. Following others [47,48], preliminary conditions to perform multiple regression analyses were generally met: the sample size was >100; the number of predictors x 10 should not be greater than sample size (here: 4 × 10 = 40 < 2374); predictors should sufficiently explain the dependent variable (Rs and R^2^′s); and the Durbin–Watson coefficient should be between 1.5 and 2.5, indicating that the residuals of the predictors were independent of each other. Last, the variance inflation factors (VIF) to test multicollinearity should be 1 < VIF < 10. 

Last, we performed two equations to calculate the direct and indirect effects. In the first equation, we calculated the direct effect of past NSSIs on current sleep complaints, via current NSSIs. In the second equation, we calculated the direct effect of past NSSIs on current NSSIs via current sleep complaints. 

The level of significance was set at alpha < 0.05. All statistical computations were performed with SPSS^®^ 28.0 (IBM Corporation, Armonk, NY, USA) for Apple Mac^®^.

## 3. Results

### 3.1. General Information

Table 1 provides the descriptive and inferential statistical overview of sociodemographic information, presented separately for female and male participants. 

Regarding the educational levels, compared to males, females had higher degrees (high school; Bachelor’s; Master’s; PhD).

Regarding the civil status, compared to males, females were more often singletons. Compared to females, males were more often married or widowed. 

With regards to employment, compared to males, females were more often unemployed, but also more often students. Compared to females, males were less often unemployed, and less often students.

Regarding suicide attempts, compared to males, females reported less often a suicide attempt. 

Concerning nightmares, compared to males, females reported less often to experience nightmares.

### 3.2. Correlations between Age, Current Sleep Patterns and Nightmares, and Current and Non-Suicidal Self-Injury 

Table 2 provides the overview of the correlation coefficients and the descriptive statistical indices.

Age was modestly related to current sleep complaints, nightmares, and past and current NSSI.

Higher current sleep complaints (PSQI and ISI) were associated with past and current NSSI, and with more nightmares in the last six months.

Current NSSIs were associated with past NSSIs and with more nightmares in the last six months. Note that the correlation coefficient of r = 0.52 between current and past NSSIs implies that the variance of the past self-injury behavior explained 27.2% of the variance of the current self-injury behavior (R^2^ = (0.52)^2^ = 0.272), or the other way around: 72.8% of the variance of the current self-injury behavior remained unexplained. 

### 3.3. Current Sleep Problems and Past and Current Non-Suicidal Self-Injury between Female and Male Participants

Table 3 provides the descriptive and inferential statistical indices of past and current self-injury behaviors and current sleep problems between female and male participants. 

Male and female participants did not differ with regards to age, sleep disturbances, and current NSSIs (trivial to small effect sizes). Compared to their male counterparts, female participants reported more past NSSIs. 

### 3.4. Correlations between Past Non-Suicidal Self-Injury and Occurrence of Suicide Attempts

The eta correlation coefficient was 0.297; higher past NSSIs predicted the occurrence of suicide attempts within the last 12 months.

### 3.5. Predicting Current Non-Suicidal Self-Injury 

Table 4 provides the statistical overview of the multiple regression equation to predict current non-suicidal self-injury.

More pronounced past NSSIs and more nightmares predicted higher current NSSIs; current sleep complaints (PSQI; ISI) were excluded from the equation, as they did not reach statistical significance. The predictive power of the predictors was moderate to high (R^2^ = 0.366 = 36.6%).

### 3.6. Predicting Insomnia (Insomnia Severity Index; ISI)

Table 5 provides the statistical overview of the multiple regression equation to predict insomnia.

More pronounced past and current non-suicidal self-injury and female gender predicted higher insomnia scores; age was excluded from the equation, as it did not reach statistical significance. The predictive power of the predictors was modest (R^2^ = 0.062 = 6.3%)**.**

### 3.7. Predicting Sleep Complaints (Pittsburgh Sleep Quality Index; PSQI)

Table 6 provides the statistical overview of the multiple regression equation to predict sleep complaints.

More pronounced current self-injury behaviors and female gender predicted higher sleep disturbances scores; age and past self-injury behaviors were excluded from the equation, as these dimensions did not reach statistical significance. The predictive power of the predictors was modest (R^2^ = 0.064 = 6.4%).

### 3.8. Direct and Indirect Effects of Past Self-Injury Behavior on Sleep Disturbances, Via Current Self-Injury Behaviors

To calculate the direct and indirect effects of past self-injury behavior on sleep disturbances via current non-suicidal self-injury, we followed Rudolf and Müller [49] and Abdoli et al. [50]. Table 7 reports the equation.

As shown in Table 7, both the direct and indirect effects of past non-suicidal self-injury on current sleep disturbances (PSQI) were modest. 

### 3.9. Direct and Indirect Effects of Past Non-Suicidal Self-Injury on Current Non-Suicidal Self-Injury, Via Current Sleep Disturbances (PSQI)

To calculate the direct and indirect effects of past non-suicidal self-injury on current non-suicidal self-injury via sleep disturbances, we again followed Rudolf and Müller [49] and Abdoli et al. [50]. Table 8 reports the equation.

As shown in Table 8, both the direct and indirect effects of past non-suicidal self-injury on current non-suicidal self-injury were high, whereas the contribution of current sleep disturbances (PSQI) on current non-suicidal self-injury was modest. 

## 4. Discussion

The key findings of the present study on non-suicidal self-injury (NSSI) behavior among a larger sample of young adults were as follows: first, past NSSIs predicted current NSSIs. Second, current sleep patterns had a modest impact on the association between past and current NSSIs. Third, compared to male participants, female participants did not report more sleep complaints or more current NSSIs, but did report more past NSSIs. Fourth, past NSSIs predicted the occurrences of nightmares and suicide attempts. The present results expand upon previous results in the following four ways. First, while previous studies focused on the association and influence of sleep on NSSIs, here, in a quasi-longitudinal design, we showed that current sleep patterns were modestly associated with current NSSIs, when past NSSIs were introduced as a further factor. Second, data were extracted from a larger non-clinical sample of young adults, for whom, by definition, the pressure to achieve on the economic and mating markets is particularly high. Third, against expectations, compared to male participants, female participants neither reported more sleep issues, nor more current NSSIs, but did report more NSSIs twelve months ago. Fourth, the occurrence of nightmares appeared to be important. Overall, the present pattern of results supports the notion of past and current NSSIs, poor sleep, nightmares, and suicide attempts as proxies of distressed mental health and deteriorated emotion regulation. 

Five hypotheses and five research questions were formulated, and each of these is considered now in turn. 

First, following others [25] we predicted that poor sleep and past and current NSSIs were associated, and data did support this assumption. However, as shown in Table 2, correlation coefficients were small to medium, suggesting that further unassessed, and latent factors might have had a stronger impact on such associations (see also below). The novelty of the present results is three-fold: first, data were carried out from a larger and non-clinical sample of young adults. Second, we introduced the recall of NSSIs occurred twelve months ago. Third, while in previous longitudinal studies [26,27,28,29] sleep patterns were considered as predictors and NSSIs as dependent variable, here, the study design was such to introduce recalled past NSSI as predictor, and sleep patterns as dependent variables. As such, we claim that this methodological change of paradigm expanded upon the current knowledge in the field in an important fashion.

With the second hypothesis we assumed that younger age was associated with higher NSSIs, and data did confirm this. Thus, the present results replicated and confirmed what has already been reported elsewhere [15,19]. The quality of the data does not allow a deeper understanding of this association. We know from imaging and neurophysiological studies on morphological brain changes among adolescents and individuals crossing into (emerging) adulthood that younger age appeared to be associated with higher impulsivity [51,52,53,54]. In the same vein, impulsive and compulsive symptoms become apparent during young adulthood, which, by definition, is a critical time for brain development and the establishments of life goals [55]. From late childhood to young adulthood, brain maturation occurs in brain regions associated with cognitive control and goal-directional behavior, including working memory, social cognition, and inhibitory control [56]. Not surprising, a bidirectional relationship between impulsivity and NSSIs was observed among 782 adults within a time lapse of three years [57]. To conclude, we assume that also in the present study ongoing brain maturation, impulsivity and higher non-suicidal self-injury patterns might be associated with younger age. On the flip side (see Table 2) correlation coefficients between age and past and current NSSIs were modest and trivial. As such, the influence of age should not be overestimated. This statement received further support from developmental studies, in which cognitive control had a larger importance on risk-taking and impulsivity, compared to age [58].

With the third hypothesis, we assumed that, compared to their male counterparts, female participants would report more sleep-related issues, though, data did not support this assumption (see Table 3). As such, the present results do not match those of previous studies [34,35,59,60]. Again, the quality of the present data does not allow a thorough explanation of this zero-result. One explanation might be that other studies did fully rely on *p*-values, which, by definition, are sensitive to sample sizes. In contrast, as shown in Table 3, we relied on effects sizes; as such, effect sizes are more precise to accurately reflect the correct mean differences [61].

With the fourth hypothesis we assumed that compared to male participants, female participants would report more NSSIs, although this assumption was only partially confirmed. More specifically, the assumption was confirmed about recalled and past NSSIs, but not about current NSSIs (Table 3). As such, the present results do not match what has been observed before [12,19]. To explain this mismatch between the present and previous results, again, the quality of the data does not provide more insight. We advance the following, although admittedly highly speculative assumptions: (1) compared to previous studies the results of the current study were drawn from a larger non-clinical sample of young adults, leading thus to different results; (2) data on past NSSIs have not been investigated so far, thus, comparisons might be biased; (3) different measurements to assess NSSIs were used, which in turn might have yielded incongruent overall results.

With the fifth and last hypothesis, we assumed that NSSIs would be associated with the occurrence of nightmares, and data did confirm this. As such, we were able to replicate what has been observed before [32,33]. We claim that this pattern of results is in accord with the assumption that in individuals with NSSIs, the underlying cognitive-emotional information processing appears to reflect a highly distressed psychobiological system and a dysregulated emotion processing. 

Next, we also formulated five exploratory research questions. 

With the first exploratory question we investigated if NSSIs 12 months ago could predict the occurrence of nightmares, and the answer was yes. As such, this pattern of result is novel, and adds to the current knowledge on the relationship between NSSIs, nightmares, and sleep in a new and important fashion. To explain the possible underlying mechanisms, we refer to the more general discussion below.

Second, we investigated if past NSSIs predicted current NSSIs, and the answer was again yes: past NSSIs predicted current NSSIs. In our opinion, for the following four reasons, this is the most important result of the present study. First, unlike previous studies on the relationship between NSSI and sleep, here, past NSSIs were introduced as independent predictor. Therefore, second, this study design changed the paradigm in this field or research. Third, it appeared that the non-suicidal self-injury-related behavioral pattern remained quite stable over time. However, fourth, we also note that the correlation coefficient was r = 0.552, or conversely: the variance of past NSSIs explained approximately 30% of the variance of the current NSSIs. Thus, approximately 70% of the variance of the current NSSI remained unexplained. One might question about the quality of the present quasi-longitudinal study design, and about participants’ ability to recall their non-suicidal self-injury-related behavior 12 months ago. This objection is justified; however, in order to minimizing the recall bias, we asked explicitly participants to vividly imaging what they were doing 12 months previously. From studies on mood induction via memory recall we know that this technique appears reliable to retrieve specific information from the long-term memory [62,63].

The third research question asked was if past NSSI-behavior could predict suicide attempts within the following 12 months, and the answer was yes. As such, in our opinion, this result underscores and justifies the present quasi-longitudinal study design. The result also confirms that NSSIs are not limited to future NSSIs (see also results of the second research question), but to an increased risk to turn NSSIs into suicide attempts [2,64].

The fourth research question asked was if past NSSIs predicted current poor sleep, and the answer was again not as straightforward as expected. While the correlation coefficients were trivial to small (Table 2), a series of regression equations revealed that both past and current NSSIs predicted current poor sleep. As such, again, we claim that the quasi-longitudinal study design allowed to gain further insight into the psychological mechanisms of NSSIs and poor sleep. 

Fifth, and last, we explored, which factors could predict current sleep quality (Pittsburgh Sleep Quality Index [PSQI], Insomnia Severity Index [ISI]) and current NSSIs. Note that to answer to these questions, we performed a series of equations to test the direct and indirect effects of possible predictors. Briefly, both past and current NSSI predicted poor sleep both directly and indirectly, while poor sleep had no predictive power on current NSSIs. This pattern of results was unexpected, as it does not match previous concepts [25,29]. However, as has already been mentioned, previous studies were either cross-sectional, and as such unable to allow to draw causal effects, or previous studies were longitudinal, but with sleep as predictor and NSSIs as dependent variables. Overall, we claim once again that with the present quasi-longitudinal study design and with past NSSIs as predictors, the entire pattern of the relationship between NSSIs and sleep changed.

Furthermore, the quality of the present study does not allow a deeper understanding of the cognitive-emotional and neurophysiological mechanisms to explain the associations between past and current NSSIs and poor sleep patterns. However, to counterbalance, we refer to previous findings in the field of cognitive-emotional sleep regulation and NSSIs. 

First, as mentioned above, we know from imaging and neurophysiological studies on morphological brain changes among adolescents and individuals crossing into (emerging) adulthood [51,52,53,54] that impulsive and compulsive symptoms become apparent during young adulthood, which, by definition, is a critical time for brain development, for the establishments of life goals [55], and for successful mating [8,9,10]. Above all, regarding the associations between suicide attempts and successful mating, previous results evidenced that higher risks of suicidal behavior were associated with less success in mating and with low sexual activities within the last 12 months [65,66,67]. Next, from late childhood to young adulthood, brain maturation occurs in brain regions associated with cognitive control and goal-directional behavior, including working memory, social cognition, and inhibitory control [56], with a bidirectional relationship between impulsivity and NSSIs [57]. We also note that cognitive control had a larger importance on risk-taking and impulsivity, compared to age [58]. As such, we claim that also in the present study ongoing brain maturation, impulsivity and higher non-suicidal self-injury patterns were associated in a bi-directional fashion.

Second, there is sufficient evidence that poor sleep, poor executive control, higher impulsivity, and emotion dysregulation are associated. More specifically, and as more extensively described in Khazaie et al. [25], briefly, four theoretical frameworks are proposed; these frameworks are highly intertwined, and presented separately for clarity and methodological reasons.

First, the concept of cognitive-emotional hyperarousal of poor sleep [68,69] asserts that impaired sleep is the result of dysfunctional and over-aroused cognitive-emotional processes.

Second, the psychophysiological hyperarousal model of Riemann et al. [70] expands upon the purely cognitive-emotional model in which severe sleep disturbances and depression are the endpoint of a bi-directional and deteriorating process between dysfunctional cognitive-emotional processes and sleep-impairing physiological changes.

Third, the cortical hyperarousal model of Fernandez-Mendoza et al. [71] and Zhao et al. [72] claims that an increased activity in brain network configurations appeared to be involved in the pathophysiology of insomnia.

Finally, fourth, following Yoo et al. [73] individuals with insomnia showed an unstable and disrupted interplay and connection between the medial-prefrontal cortex (MPFC) and its inhibitory and top-down control of the amygdala and adjacent mesolimbic structures, leading to less functional emotional responses and to a lower impulse control inhibition.

Overall, we claim that the results on brain maturation, inhibition control, and impaired sleep due to dysfunctional cognitive-emotional and physiological processes and increased NSSIs appear to reflect the same neurophysiological and cognitive-emotional processes. 

The novelty of the results should be balanced against the following limitations. First, given the large sample size, we relied on effect size calculations, as effect sizes reflect the true mean differences, while *p*-values become “significant” with increasing sample sizes. Second, the study design was quasi-longitudinal; one may claim that completing questionnaire items on the present and past NSSIs might lead to biased results. While we cannot fully rule out this risk, the following points should be considered. First, as mentioned, participants were encouraged to recall their life 12 months ago, and there is evidence that such interventions impact on the retrieval of long-term memory information [62,63]. Second, if reporting current NSSIs and recalling past NSSIs were identical or biased, then one would expect a definitively higher correlation coefficient than just r = 0.552. Nevertheless, we are aware that the present quasi-longitudinal study design is unable to replace longitudinal studies with two or more timepoints. 

Third, by definition, it is conceivable that latent and unassessed cognitive-emotional processes might have biased two of more dimensions in the same or opposite directions. As described extensively in the Introduction section, NSSIs are related to a broad variety of psychological, psychiatric, social-interactional, and cultural factors. More specifically, one might claim that symptoms of depression might have biased the whole pattern of results. While with the present data we cannot full rule out this claim, we also note that among a larger sample of adolescents the association between suicidal behavior and poor sleep was above and beyond symptoms of depression [31]. As such, it is highly conceivable that also in the present study, symptoms of depression were not confounders. 

In this view, fourth, there is sufficient evidence that sleep complaints among adolescents and young adults might be related to excessive exposure to screens (e.g., tablets, smart phones). [74,75,76,77]. As such, it is conceivable that exposure to screen might have biased the present pattern of results.

## 5. Conclusions

Among a larger sample of young adults, non-suicidal self-injury-patterns 12 months ago predicted current non-suicidal self-injury-behavior and current poor sleep. Current poor sleep was modestly associated with current non-suicidal self-injury-behavior. The quasi-longitudinal study design allowed to understand current NSSIs as the result of previous NSSIs. This statement holds also true for nightmares and suicide attempts. 

## Figures and Tables

**Table 1 ijerph-18-13011-t001:** Descriptive and inferential statistical overview of sociodemographic information, presented separately for female and male participants.

Variables	Groups		Statistics
	Female	male	
N	M (SD)	M (SD)	
	941	1433	
Age	27.16 (5.24)	27.85 (5.28)	*t*(2372) = 3.13 **, d = 0.17 [T]
	*n*/*n*/*n*/*n*	*n*/*n*/*n*/*n*	
Education (compulsory school, high school diploma, Bachelor’s, Master’s, and PhD)	166/279/328/168	240/648/436/109	X^2^(N = 2374, df = 3) = 90.11 ***
Civil status (single, married, widowed)	389/479/73	321/988/124	X^2^(N = 2374, df = 2) = 98.59 ***
Employment (unemployed, student, employed)	246/330/365	175/332/926	X^2^(N = 2374, df = 2) = 160.69 ***
Attempted suicide (yes, no)	279/662	500/933	X^2^(N = 2374, df = 1) = 7.08 **
Nightmares in the past six months (never, rarely, sometimes, lots of nights, most nights, almost every night)	256/298/252/81/20/34	208/529/437/160/54/45	X^2^(N = 2374, df = 5) = 62.95 ***

Notes: T = trivial effect size; ** = *p* < 0.01; *** = *p* < 0.001.

**Table 2 ijerph-18-13011-t002:** Descriptive and correlational statistical indices between age, sleep complaints and past and current non-suicidal self-injury.

	N	M	SD	Age	PSQI	ISI	Nightmares in the Past 6 Months	Current Self-Injury	Past Self- Injury
Age	2374	27.58	5.27	-					
Sleep disturbances (PSQI)	2374	5.68	2.66	0.000	-				
Insomnia (ISI)	2374	6.19	4.32	−0.035	0.621 **	-			
Nightmares in the past 6 months	2374	2.52	1.19	−0.062 **	0.453 **	0.476 **	-		
Current non-suicidal self-injury	2373	29.82	4.73	0.120 **	−0.238 **	−0.188 **	−0.397 **	-	
Past non-suicidal self-injury	2374	26.84	5.27	0.037	−0.150 **	−0.012	−0.290 **	0.552 **	-

Notes: PSQI = Pittsburgh Sleep Quality Index; ISI = Insomnia Severity Index; M = mean; SD = standard deviation; ** = *p* < 0.01.

**Table 3 ijerph-18-13011-t003:** Descriptive and statistical indices of age, sleep disturbances (Pittsburgh Sleep Quality Index; Insomnia Severity Index), and current and past self-injury between male and female participants.

	Gender		Statistics	
	Male	Female	*t*-test	Effect sizes
N	1433	941		
	M (SD)	M (SD)		Cohen’s d
Age	27.85 (5.28)	27.16 (5.24)	*t*(2372) = −3.13 **	0.131 [T]
Sleep disturbances (PSQI)	5.66 (2.46)	5.71 (2.95)	*t*(2372) = 0.63	0.020 [T]
Insomnia (ISI)	5.82 (3.95)	6.75 (4.78)	*t*(2372) = 5.14 ***	0.216 [S]
Current non-suicidal self-injury	28.95 (4.84)	31.15 (4.22)	*t*(2372) = 11.37 ***	0.477 [S]
Past non-suicidal self-injury	25.50 (4.88)	28.90 (5.19)	*t*(2372) = 16.18 ***	0.679 [M]

Notes: ** = *p* < 0.01; *** = *p* < 0.001; T = trivial effect size; S = small effect size; M = medium effect size.

**Table 4 ijerph-18-13011-t004:** Dimensions to predict current non-suicidal self-injury: multiple regression analysis.

Dimension	Variables	Coefficient	Standard Error	Coefficient β	*t*	*p*	R	R^2^	Durbin-Watson	VIF
Current non-suicidal self-injury	Constant	20.938	0.495	-	42.28	0.000	0.605	0.366	1.77	
	Past non-suicidal self-injury	−0.427	0.015	0.477	27.893	0.000				1.10
	Nightmares	−1.025	0.068	−0.259	15.144	0.000				1.10
		Excluded variable: Sleep complaints (PSQI); Insomnia (ISI); *t*s < 1.44, *p* > 0.20								

Notes: VIF = Variance Inflation Factor; higher scores of current and past non-suicidal self-injury reflect *lower* scores of non-suicidal self-injury.

**Table 5 ijerph-18-13011-t005:** Dimensions to predict insomnia scores (ISI): multiple regression analysis.

Dimension	Variables	Coefficient	Standard Error	Coefficient β	*t*	*p*	R	R^2^	Durbin-Watson	VIF
Insomnia	Constant	13.397	0.820	-	16.334	0.000	0.253	0.063	1.456	
	Current s non-suicidal self-injury	−0.249	0.022	−0.272	−11.284	0.000				1.468
	Past non-suicidal self-injury	−0.078	0.020	−0.095	3.855	0.000				1.522
	Gender	−1.213	0.186	−0.137	−6.506	0.000				1.125
		Excluded variable: Age: *p* > 0.87								

Notes: VIF = Variance Inflation Factor; Gender: 1 = females; 2 = males. Higher scores of current and non-suicidal self-injury reflect *lower* scores of non-suicidal self-injury.

**Table 6 ijerph-18-13011-t006:** Dimensions to predict sleep disturbances scores (PSQI = Pittsburgh Sleep Quality Index); multiple regression analysis.

Dimension	Variables	Coefficient	Standard Error	Coefficient β	*t*	*p*	R	R^2^	Durbin-Watson	VIF
Sleep disturbances	Constant	10.479	0.506	-	20.725	0.000	0.252	0.064	1.955	
	Current non-suicidal self-injury	−0.132	0.014	−0.235	−9.750	0.000				1.468
	Gender	−0.435	0.115	−0.080	−3.784	0.000				1.125
	Excluded variables: Age and past non-suicidal self-injury: *p*s > 0.09.									

Notes: VIF = Variance Inflation Factor; Gender: 1 = females; 2 = males. Higher scores of current non-suicidal self-injury reflect *lower* scores of non-suicidal self-injury.

**Table 7 ijerph-18-13011-t007:** Equation model to calculate the direct and indirect effects of past non-suicidal self-injury behavior on sleep disturbances.

*R_NSSI_P_-PSQI_*	=	Direct Effect NSSI_P_ on PSQI β	+	Indirect Effect of NSSI_P_ via NSSI_C_ βNSSI_P_-NSSI_C_ × rNSSI_C_-PSQI
r = −0.15	=	−0.095	+	0.552 × (−0.238)

Notes: PSQI = Pittsburgh Sleep Quality Index = sleep disturbances; NSSI_P_ = past non-suicidal self-injury; NSSI_C_ = current non-suicidal self-injury.

**Table 8 ijerph-18-13011-t008:** Equation model to calculate the direct and indirect effects of past non-suicidal self-injury behavior on current non-suicidal self-injury behavior.

rNSSI_P_-NSSI_C_	=	Direct Effect NSSI_P_ on NSSI_C_ β	+	Indirect Effect of NSSI_P_ via Sleep Disturbances (PSQI) βNSSI_P_-PSQI × rNSSI_C_-PSQI
r = 0.552	=	0.546	+	−0.026 × (−0.238)

Notes: PSQI = Pittsburgh Sleep Quality Index = sleep disturbances; NSSI_P_ = past non-suicidal self-injury; NSSI_C_ = current non-suicidal self-injury.

## Data Availability

Data are made available upon request to expert scientists in the field and upon clear and reasonable hypotheses.

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
