# Peer review of "When Non-Suicidal Self-Injury Predicts Non-Suicidal Self-Injury and Poor Sleep—Results from a Larger Cross-Sectional and Quasi-Longitudinal Study"

_ijerph, 2021, doi:10.3390/ijerph182413011_

Round 1

Reviewer 1 Report

The manuscript addressed a significant issue. However, the article is not nevelty.
We already know that non-suicidal self-injury experiences and frequency are related to self-injury.
Authors need to derive results related to self-injury by adding various variables in addition to non-suicidal self-injury.

Author Response

We thank Reviewer #1 for the valuable comments and for helping us to improve the quality of the manuscript. Please find the detailed point-by-point-response attached as a separate file. 

Again, thank you very much for all your kind efforts.

Reviewer 2 Report

This study confirms some existing hypotheses around the relationship between NSSI and poor sleep, and challenges others. The Authors claim their methodology and approach is different to previous work and therefore provides a new and relevant perspective on the NSSI – sleep relationship. I would agree with this observation.

For Hypotheses 2, 3, and 4 in particular, limitations in the data prevent further unpacking of the results, which points to the need for more robust longitudinal study designs in this area.

The Authors do a good job of describing the limitations of the methods and results. As such, I have no major criticisms of this paper, but do have a few suggestions for inclusion that I feel could improve the work:

Association is not causation. Ideally, a research methodology covering the relationship between NSSI and sleep would support causal inference, but none have successfully done so to date (to my knowledge).  I think this should be made more explicit than it currently is in this paper. Merely sleeping longer may not address the underlying psychological, psychiatric, social or other drivers of NSSI. The extent to which additional/better sleep might moderate NSSI prevalence of persons with underlying mental health problems and social issues is to be determined and would likely differ depending on the combination of factors experienced by any individual.

On page 4 of the Discussion the Author’s describe four overlapping theoretical frameworks. These cover the concept of hyper-arousal. I wonder if this section could include the observation from recent studies that people (younger persons unparticular) in the post-smartphone era are increasingly influenced by screen/internet/social media addiction which leads to habitual late-night screen time. This nighttime exposure to blue-light stimulus turns brains into wake-state at times when deep sleep would be natural. This ongoing disruption to circadian rhythms has meaningful impacts on mood, emotional regulation, brain development, etc…., and may be part of the unmeasured influence on sleep and NSSI that you refer to.

Though the standard of written English is high, there are several typos and small grammatical errors throughout.

Author Response

We thank Reviewer #2 for the valuable comments and for helping us to improve the quality of the manuscript. Please find the detailed point-by-point-response attached as a separate file. 

Again, thank you very much for all your kind efforts.

Round 2

Reviewer 1 Report

I believe the manuscript has been sufficiently improved to warrant publication in IJERPH.

This manuscript is a resubmission of an earlier submission. The following is a list of the peer review reports and author responses from that submission.